# The Adherence of Digital Templating of Cemented Bicondylar Total Knee Arthroplasty Reveals Gender Differences

**DOI:** 10.3390/jcm12031079

**Published:** 2023-01-30

**Authors:** Julian Koettnitz, Jara Tigges, Christian Dominik Peterlein, Matthias Trost, Christian Götze

**Affiliations:** 1Department of General Orthopaedics, Auguste-Viktoria-Clinic Bad Oeynhausen, University Hospital of RUB-Bochum, Am Kokturkanal, 32545 Bad Oeynhausen, Germany; 2Department of Orthopaedics and Traumatology, St. Josef-Hospital, Ruhr University Bochum, 44791 Bochum, Germany

**Keywords:** digital templating, gender differences, total knee arthroplasty

## Abstract

Introduction: Preoperative digital templating is a standard procedure that should help the operating surgeon to perform an accurate intraoperative procedure. To date, a detailed view considering gender differences in templating total knee arthroplasty (TKA), stage of arthrosis, and the surgeons’ experience altogether has not been conducted. Methods: A series of 521 patients who underwent bicondylar total knee arthroplasty was analyzed retrospectively for the planning adherence of digital templating in relation to sex, surgeon experience, and stage of arthrosis. Pre- and postoperative X-rays were comparably investigated for planned and implanted total knee arthroplasties. Digital templating was carried out through mediCAD version 6.5.06 (Hectec GmbH, 84032 Altdorf, Germany). For statistical analyses, IBM SPSS version 28 (IBM, 10504 Armonk, NY, US) was used. Results: The general planning adherence was 46.3% for the femur and 41.8% for the tibia. The Mann–Whitney U test revealed a gender difference for templating the femur (z = −5.486; *p ≤* 0.001) and tibia (z = −3.139; *p* = 0.002). The surgeon’s experience did not show a significant difference through the Kruskal–Wallis test in the femur (K–W H = 4.123; *p* = 0.127) and the tibia (K–W H = 2.455; *p* = 0.293). The stage of arthrosis only revealed a significant difference in the planning of the femur (K–L-score (K–W H = 6.516; *p* = 0.038) alone. Discussion/Conclusion: Digital templating for total knee arthroplasty brought up gender differences, with oversized implants for women and undersized implants for men. A high stage of femoral arthrosis can lead to the under and oversized planning of the surgeon. Since the surgeon’s experience in planning did not show an effect on the adherence to templating, the beneficial effect of digital templating before surgery should be discussed.

## 1. Introduction

Worldwide, total knee arthroplasty is a popular surgical intervention [1] as a positive outcome can restore a painless full range of motion. For a successful surgery, thorough preparation must be carried out, which includes appropriate planning of the implant [2,3]. Planning the total knee arthroplasty can be performed digitally via an X-ray or a 3D model. In any case, an oversizing of the implants, especially the femoral component, should be avoided [4]. Otherwise, complications such as postoperative chronic pain or limitations on mobility could be possible. Still, there is only a postoperative satisfaction of around 80% after total knee arthroplasty, which could be due to insufficient preoperative planning [5]. Preoperative planning not only simplifies the determination of the exact implant size, but it can also improve the surgery procedure and postoperative range of motion and help to restore patient-specific biomechanics [6,7]. Many studies in recent years have focused on planning adherence in total hip arthroplasty (THA) [6,8,9]. For example, Dammerer et al., 2022 showed a sex difference in templating the femur stem, with better results for women. Likewise, Luger et al., 2022 revealed gender differences in planning adherence during total hip arthroplasty. Regarding surgeon experience and digital planning, several studies were conducted for THA. Some studies demonstrated differences, with better planning adherence for more experienced surgeons, while others could not find any differences [10,11,12]. An additional detailed view of the stage of arthrosis and its outcome for digital planning adherence has not been conducted so far and remains unclear. The study of gender differences and surgeon experience in digital templating for THA is well known. However, there is only a little information about gender differences for total knee arthroplasty. To the best of our knowledge, there is no survey considering the sex differences, surgeons’ experience, and stage of arthrosis in planning adherence altogether for bicondylar total knee arthroplasty. In this study, an investigation of planning adherence in relation to sex, the experience of the surgeon, and the degree of arthrosis for bicondylar total knee arthroplasty was conducted. Thereby, the exact planning and over and under-planned sizes were analyzed in relation to the gender, experience of the surgeon, and the degree of arthrosis. The aim of the study was the detection of repetitive significant differences in the planning adherence regarding gender, surgeons’ experience, or stage of arthrosis to prevent possible failures from occurring and to improve the clinical work process.

## 2. Methods

### 2.1. Study Design

The present study was performed according to the Strengthening the Reporting of Observational Studies in Epidemiology (STROBE) [13]. This study was conducted in the Department of Orthopedic Surgery of a University Hospital. A series of 521 patients (mean age 70.37 years ± 9.20) from 2018 to 2021 at the hospital was analyzed. The data were collected within 6 months in the year 2022 and entered into Excel^®^ (Microsoft Corp., 98000-98099 Redmond, WA, US) in consecutive order. The data were collected from the digital patient documentation system, the digital patient archive system, and the digital radiography system and were recorded metrically, ordinally, and nominally.

Patients undergoing primary cemented bicondylar total knee arthroplasty were examined and their eligibility to participate in this study was assessed. The inclusion criteria were (1) patients with primary and secondary arthrosis undergoing a cemented primary knee arthroplasty, (2) cemented primary bicondylar total knee arthroplasty, (3) accessible patient data, (4) implantation of S&N Genesis II CR/PS, Journey II BCS/CR, and (5) TKA: anteromedial approach (with median skin incision). The exclusion criteria were (1) revision knee arthroplasty, (2) uncemented primary knee arthroplasty, (3) inaccessible patient data, and (4) a different approach than anteromedial. The present study was approved and registered by the ethics committee (HDZ-NRW) of the RUB University of Bochum and conducted according to the principles expressed in the Declaration of Helsinki. For an overview, see Figure 1 (workflow).

### 2.2. Preoperative X-ray Routine

A standardized preoperative digital radiograph of the anterior-posterior view of the full leg in a standing position (with a view of the hip and ankle joint), a lateral view of the knee, and a patella defile view were taken of the patients. A standardized metallic radiopaque ball with a diameter of 25 mm was used as a reference for determining the magnification factor.

### 2.3. Digital Templating

Digital templating was carried out with mediCAD^®^ version 6.5.06 (Hectec GmbH, 84032 Altdorf, Germany ) in a standardized manner. First, the full leg stand view was used for planning. Afterward, the lateral scan was used for planning purposes. Initially, scaling at 25 mm was performed automatically by the software with the metallic radiopaque ball as the reference. Then, the correct operation side was marked in the software which unlocks the button for the implant types and sizes. After choosing the correct implant type, the implant size was selected by the surgeon. For the full leg stand view, implants in an anterior-posterior view were planned and for the lateral view, implants in a mediolateral view were planned. For the femoral side, the position was set at a beta angle of 5, 6, or 7 degrees. For the tibia, the correct position in line with the ankle joint was important. In general, the digital template was adjusted to the size in the X-ray and should not exceed the size shown in the X-ray. Once the tibia and femur had been planned, the image was saved and transferred to the digital radiology system.

### 2.4. Outcome of Interest

The age, sex, side, body mass index (BMI), length of hospital stay, time of operation, type of implants, size of the implants pre- and postoperative, the experience of surgeons (classified through the quotient of the number of surgeries from the particular surgeon divided by the number of surgeries from the surgeon with the highest amount of TKA), and grade of arthrosis (Kellgren–Lawrence score) were recorded. Data analysis between sex, the experience of the surgeons, the Kellgren–Lawrence score, and the difference between the planned and implemented size were performed.

### 2.5. Statistical Analyses

All statistical analyses were performed using the software IBM SPSS version 28 (IBM, 10504 Armonk, NY, US). The significance level was set two-sided with α = 0.05. Age, BMI, length of hospitalization, length of intensive care stay, pre- and postoperative laboratory parameters, and frequency of transfusion were listed metrically. Sex, side, preconditions, and systemic and surgical complications were listed nominally. For patients, demographic data were analyzed by the mean, standard deviation, and percentage. For general planning adherence, the planned size of the femur and tibia was subtracted from the implanted size. The exact planning adherence was 0. For planning adherence and sex, experience, and the degree of arthrosis, the nominal, dichotomous data were analyzed by Fischer’s exact test. This test was used because of its independence from the sample size. The additional effect size (small .10; medium .30; and large .50) was used to detect the clinical impact of a significant result. For planning adherence and sex, each number from femur +III to −II was analyzed with sex. In addition, the exact match, above, and under sizes from the femur and tibia were listed ordinally and analyzed through the Mann–Whitney U test to detect differences between sizes and sex and support Fischer’s exact test. The z-values of the Mann–Whitney U test reflect the standard deviations. A z-value > −2 or 2 indicates that measured significances cannot be explained by theoretical random patterns. For planning, adherence, experience, and degree of arthrosis, each experience level or degree of arthrosis was analyzed with Fischer’s exact test. To support the results, the Kruskal–Wallis test was used because it enables the evaluation of more than three independent samples. The asymptomatic p and Kruskal–Wallis H values were used with regard to the significance level. Kruskal–Wallis H values above 5.99 and asymptomatic *p*-values < 0.05 show significance. Furthermore, linear regression with an effect strength of R^2^ (Cohen) (weak = 0.02, middle = 0.13, and high = 0.26 variance clarification) was used to control the results. Through regression analysis, it was investigated whether the degrees of experience or arthrosis have a linear influence on planning adherence.

## 3. Results

### 3.1. Recruitment Process

For this retrospective study, data from *n* = 865 patients with primary knee arthroplasties from the years 2018–2021 during the inpatient stay and follow-up were analyzed. In total, N = 344 patients with cemented and uncemented primary unicondylar knee arthroplasty and primary cemented bicondylar arthroplasty with primary cemented implants other than Journey II or Genesis II were excluded. An in-depth analysis of *n* = 521 was performed. For an overview, see Figure 2.

### 3.2. Demographics

Data from *n* = 521 patients were analyzed. Overall, 36.1% (188 of 521) of these patients were men and 63.9% (333 of 521) were women. The mean age was 70.37 (±9.2). In 96.4% (*n* = 502) of the cases, primary arthrosis was the indication, and in 2.7% of the cases, secondary arthrosis was the indication. A total of 17.5% (*n* = 91) showed a severe varus in the arthrosis, and 13.1% (*n* = 68) had a severe valgus in the arthrosis. No patient showed a type I Kellgren–Lawrence score. However, 2.3% (*n* = 12) were classified with a type II Kellgren–Lawrence score, 29.6% (*n* = 154) with a type III Kellgren–Lawrence score, and 62.4% (*n* = 325) with a type IV Kellgren–Lawrence score. According to the WHO criteria, the mean BMI was 30.95 (±6.71); 0.6% (*n* = 3) were underweight (BMI < 18.5 kg/m^2^), 13.8% (*n* = 72) were normal-weight (BMI ≥ 18.5 and ≤25 kg/m*2*), 35.5% (*n* = 185) were overweight (BMI ≥ 25 and ≤29.9), 23.0% (*n* = 120) were obese type I (BMI ≥ 30 and ≤34.9 kg/m*2*), 13.8% (*n* = 72) were obese type II (BMI ≥ 35 and ≤39.9 kg/m*2*), and 10% (*n* = 52) were obese type III (BMI ≥ 40 kg/m*2*). The mean pain symptom value before surgery was 6.4 points (±2.21) (*n* = 259) and a day before leaving the hospital after surgery was 1.57 points (±0.55) (*n* = 269). The mean ROM before surgery was 99.97 degrees (±16.26) (*n* = 509), after surgery was 88.71 degrees (±6.41) (*n* = 483), and at the time of re-appointment was 105.12 degrees (±18.32). The mean length of stay was 10.13 days (±3.64), and the mean operation time was 99 min (±0.49 min). In total, 16 surgeons were listed for total knee arthroplasty from 2018 to 2021 and 34.9% Journey II implants (*n* = 182) and 64.5% Genesis II implants (*n* = 336) were implanted.

### 3.3. General Planning Adherence

Femur: 11 of 521 cases were missing. The exact planning for the femoral side was achieved in 46.3% (*n* = 241) of the cases. In 22.8% of the cases (*n* = 119) one size bigger, in 2.5% (*n* = 13) two sizes bigger, and in 0.4% (*n* = 2) three sizes bigger were planned. In 23.4% of the cases (*n* = 122), one size smaller, and 2.5% of the cases (*n* = 13) two sizes smaller were planned.

Tibia: 126 of 521 cases were missing. The exact planning adherence of the tibia was fulfilled in 41.8% of the cases (*n* = 218). One size bigger was planned in 9.6% (*n* = 50) and two sizes bigger were planned in 1.7% (*n* = 9) of the cases. One size smaller was planned in 20.3% (*n* = 106) of the cases, and two sizes smaller were planned in 1.9% (*n* = 10). For an overview, see Table 1.

### 3.4. Planning Adherence and Sex

Femur: The exact adherence of the femur implant was achieved in 44.8% (*n* = 82) of the cases in male patients and 48.6% (*n* = 159) of the cases in female patients. Femoral plus I was implanted in 12.5% (*n* = 23) of the cases in male patients and 41.5% (*n* = 96) in female patients. Femoral plus II was used in 2.1% (*n* = 4) of the male cases and 2.8% (*n* = 9) of the female cases. Femoral plus III was implanted in 0.5% (*n* = 1) of the male cases and 0.3% (*n* = 1) of the female cases. Femur minus I was used in 37.1% (*n* = 68) of the male patients and 16.5% (*n* = 54) of the female cases. Femur minus II was found in 2.7% (*n* = 5) of the male cases and 2.44% (*n* = 8) of the female cases. Overall, there was a significant difference in implanted sizes in relation to sex (z = −5.486; *p* ≤ 0.001). For significant group differences in detail, see Table 2.

Tibia: The exact adherence of the tibia implant was achieved in 45.3% (*n* = 63) of the male cases and 60.5% of the female cases. Tibia minus I was implanted in 37.4% (*n* = 52) of the male cases and 21.0% (*n* = 54) of the female cases. Tibia minus II was used in 3.5% (*n* = 5) of the male cases and 1.9% (*n* = 5) of the female cases. Tibia plus I was found in 11.5% (*n* = 16) of the male cases and 13.2% (*n* = 34) of the female cases. Tibia plus II was implanted in 1.4% (*n* = 2) of the male patients and 2.7% (*n* = 7) of the female patients. Altogether, there was a significant difference in the implanted sizes in relation to sex (z = −3.139; *p* = 0.002)

For significant group differences in detail, see Table 3.

### 3.5. Planning Adherence and Experience

The experience of the surgeons was classified through the number of surgeries. The surgeon with the most operations was set as the reference (100%). All procedures of the reference surgeon were divided by the surgeries performed by the other surgeons. Four categories were formed: (4) > 75%, (3) < 75%, (2) < 50%, and (1) < 25%. Overall, there was no surgeon categorized in group 3. Regarding experience, the femoral side was not significantly different (K–W H = 4.123; *p* = 0.127) and the tibial side showed no significant relation to the planner’s experience (K–W H = 2.455; *p* = 0.293). In addition, the linear regression did not show a significant relation adherence to the experience of the surgeon (r = 0.15; r^2^ = 0.023; *p* = 0.555). Group 2 (<50%) revealed two significant results for femoral plus I and tibia minus II implantations against all other groups (phi = 0.107, *p* = 0.022; phi = 0.133, *p* = 0.013), and group I showed a significant result in less tibial minus II implantations against all other groups (phi = 0.105, *p* = 0.037). Detailed group differences were calculated through the exact Fisher’s test. For more details, see Table 4.

### 3.6. Planning Adherence and Degree of Arthrosis

The degree of arthrosis was classified using a Kellgren–Lawrence score of I to IV. No case was classified with a Kellgren–Lawrence I score. In total, 2.3% (*n* = 12) were classified as II, 29.6% (*n* = 154) were classified as III, and 62.4% (*n* = 325) were classified as IV. In general, the femoral side showed a significant difference between the planning adherence and the K–L score (K–W H = 6.516; *p* = 0.038), whereas the tibial side showed no significant difference (K–W H = 2.583; *p* = 0.275). The K–L-score III was significantly more related to femur plus I implantation (phi = 0.103; *p* = 0.30) against all other groups. Linear regression could not show a connection between all implantation matches to the degree of arthrosis (r = 0.128; r^2^ = 0.016; *p* = 0.820). For the femoral side, *n* = 37 cases were missing. For the tibial side, *n* = 151 cases were missing (Table 5).

## 4. Discussion

This study retrospectively analyzed the planning adherence for primary bicondylar total knee arthroplasty throughout the standard medial parapatellar approach of a single center in 521 patients. To the best of our knowledge, it is the first investigation that analyzes gender differences, differences in planning adherence in relation to the stage of arthrosis, and surgeons’ experience altogether.

Digital planning is a routine process before total knee arthroplasty. Two-dimensional digital planning has widely established itself, although patient-specific magnet resonance imaging (MRI) and computed tomograph (CT) planning methods are also shaping the market [4,14,15]. Digital templating is an underestimated crucial feature that allows the surgeon to plan the best implant for the specific patient. It starts a preoperative preparation process that makes the decision of an implant size safer and more reliable, i.e., better outcome results were shown when planning the implants before arthroplasty [16,17].

For 2D-templating, Klag et al., 2022 revealed only a 50% prediction rate for surgeons for the tibial side and a 77% prediction rate for the femoral side [16,18]. In this analysis, the general exact planning adherence was 46.3% on the femoral side and 41.8% on the tibial side. Many factors can play a role in this process. The individual experience of planning of the surgeon, images which deviate from the standard whole leg position by processing through the radiology department, or simply the “over” or “under” sized planning of the surgeon. Other studies revealed similar results for 2D planning within 30–40%, and even more recent methods showed an exact planning adherence of about 40% [3,18].

The influence of the experience of a surgeon in digital templating can be a decisive factor in the success of arthroplasty [19,20]. The hypothesis of this study that surgeons’ experience leads to different planning adherences did not reveal profound results. Interestingly, although the result was not significantly different, it should be mentioned that the less experienced surgeon group showed a more frequent exact matching in the femur and tibia (54.5% and 58.4%, respectively). This is in line with Shichman et al., 2020 who revealed an equal planning capability of fellows and residents in total hip arthroplasty [11]. These results show that surgeons’ plans have only moderate predictive power, regardless of experience. The question is if the experience of the surgeon or the technology is responsible for the results.

Digital templating and sex are discussed in the literature. Other studies could not reveal any gender differences in relation to planning [6,21]. Since there are gender differences in the outcome of arthroplasties [22,23], a precise gender, individual-conformed plan could have a positive influence. In this study, one of the hypotheses was that gender influences the planning adherence in digital templating. Luger et al., 2022 [9] detected a gender difference planning in total hip arthroplasty. They showed a gender difference in planning the offset option to the implanted arthroplasty; men were significantly planned more often with an −1 offset option whereas women were planned more often with an exact match. In this investigation, women were significantly planned more frequently with a femur size +1 (w 41% vs. m 12.5%), but men with a size −1 (m 37.1% vs. w 16.5%). The exact planning for the tibia was revealed for women to be 60.5% vs. men with 45.3%. Like the femur, men were undersized, −1 in 37.4% vs. women in 21.0% of the cases. A possible reason for undersized implants could be the tendentially bigger knees of male persons, whereby overplanning would likely be avoided. A new experimental study by Yue et al., 2022 invented an algorithm that gave a prediction accuracy of 88.23% based on gender and BMI [24].

As the degree of arthrosis for a total arthroplasty is mostly in the final stage, a comparative analysis in the literature was barely found [25]. Regardless, most patients only show up for total knee arthroplasty when pain is unmanageable, and mobility is substantially limited. This is often in line with advanced stages of arthrosis [26]. The results of this study revealed that there is no difference between Kellgren–Lawrence stages II–IV, but overall, the adherence was poor. Only for the femur side did the regression analyses reveal a significant difference (K–W H = 6.516; *p* = 0.038). Additionally, especially for extremely deformed joints, a precisely planned implant could be helpful for the surgery and might reduce the time of the operation. As a result, in the future, more 3D templating will be established. Accuracies of up to 90% and independence from different patient weights against 2D planning have been demonstrated already [27,28].

However, this study has its limitations. First, this single-center analysis could only image the planning strategy of the in-house surgeons. Second, from the 521 analyses, some planning details for the femur and tibia were missing, which lead to a reduction in the patient count in the results. Third, there is a difference in the count of men and women, which could be a bias factor. Fourth, the planner’s experience was categorized through a hospital’ s internal calculation; other hospitals could use a different categorization method. Fifth, the classification of the stage of arthrosis is dependent on the subjective opinion of the indicating surgeon and could thus lead to a distortion of the results.

## 5. Conclusions

The 2D digital planning of total knee arthroplasty shows a gender difference in the exact templating. The more frequent undersized implants for males and oversized implants for females should be noted. The surgeon’s experience does not influence the exact templating of implants. Planning the femoral component in high-stage arthrosis is more related to under and oversizing. As the general planning adherence was under 50%, it should be questioned if conventional digital planning has a deep impact on the total knee arthroplasty. There is a further need for planning tools that could help surgeons to plan more accurately. Currently, low-dose CT 3D scans, MRI 3D scans, or 3D reconstructions from 2D X-rays are in use to improve digital templating [29,30,31]. Another new technology is preoperative 3D-printed implants for planning surgery [31], which could be interesting for difficult cases. The greatest potential could be in 3D CT scans, where implants can be templated in all directions and more precisely than in the 2D technique. Nevertheless, clinics should still not lose sight of economic efficiency. At this point, digital 2D templating is still very important as a standard for most surgeons.

## Figures and Tables

**Figure 1 jcm-12-01079-f001:**
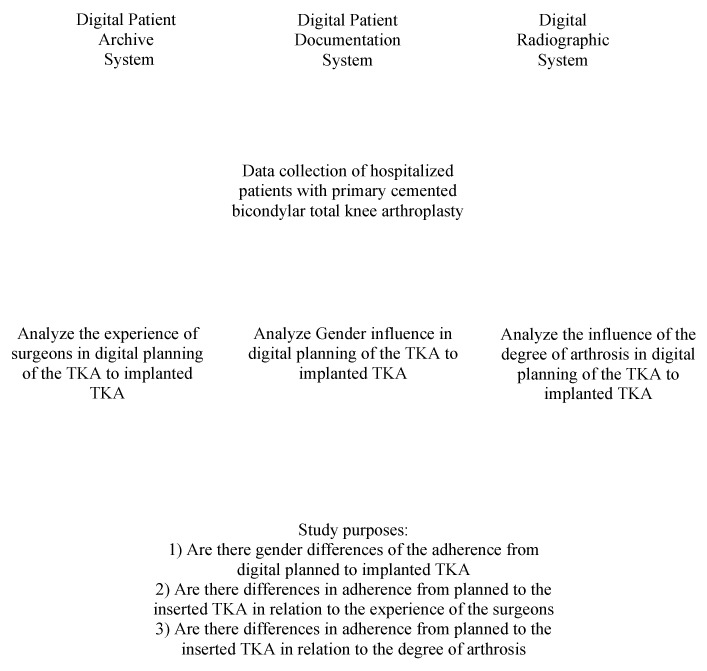
Workflow. Data acquisition was consecutive. First, the examination of cases was completed through the current patient recording system and, for control and additional data acquisition, through the archiving program of not digitally recorded documents. The examination of preoperatively planned implants was carried out through the digital X-ray image viewer. The data collection then was analyzed for the study purposes shown in the figure.

**Figure 2 jcm-12-01079-f002:**
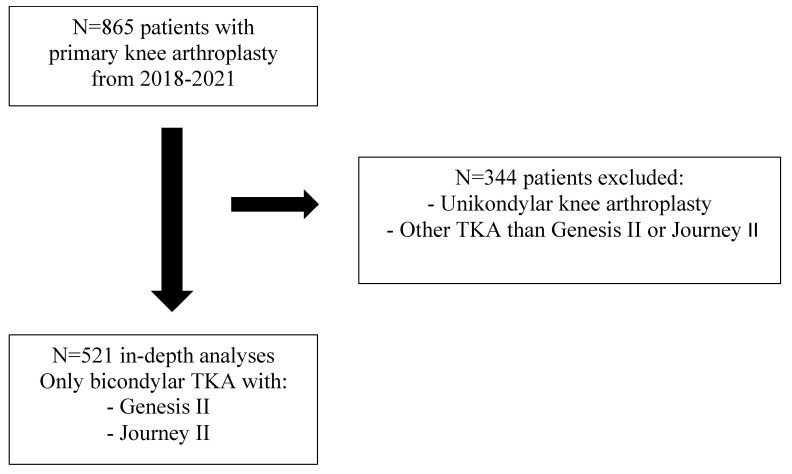
Recruitment process.

**Table 1 jcm-12-01079-t001:** General planning adherence of femur and tibia.

Sizes	Exact	+1	+2	+3	−1	−2
Femur	241 (46.3%)	119 (22.8%)	13 (2.5%)	2 (0.4%)	122 (22.8%)	13 (2.5%)
Tibia	218 (41.8%)	50 (9.6%)	9 (1.7%)	-	106 (20.3%)	10 (1.9%)

**Table 2 jcm-12-01079-t002:** Femoral planning adherence and sex; *p* (value of significance from Fischer’s exact test).

Femur (*n* = 510)	Men (*n* = 183)	Women (*n* = 327)	phi	*p*
Exact match	44.8%	48.6%	0.037	0.460
Plus I	12.5%	41.5%	0.190	<0.001
Plus II	2.1%	2.8%	0.017	0.779
Plus III	0.3%	0.5%	0.018	1.0
Minus I	37.1%	16.5%	0.232	<0.001
Minus II	2.7%	2.44%	0.009	1.0

**Table 3 jcm-12-01079-t003:** Tibial planning adherence and sex; (value of significance from Fischer’s exact test).

Tibia (*n* = 395)	Men (*n* = 139)	Women (*n* = 256)	phi	*p*
Exact match	45.3%	60.5%	0.146	0.004
Plus I	11.5%	13.2%	0.025	0.752
Plus II	1.43%	2.7%	0.041	0.503
Minus I	37.4%	21.0%	0.176	<0.001
Minus II	3.5%	1.9%	0.050	0.332

**Table 4 jcm-12-01079-t004:** Femoral and tibial planning adherence in relation to surgeon experience; *p*-values from each Fischer’s exact test. Each experience group and the match were tested against all other groups and matches. * = significantly less often implanted against all other experience groups, ** = significantly more often implanted than all other experience groups.

Femur (*n* = 509)	<25% (*n* = 119)	<50% (*n* = 152)	>75% (*n* = 238)
Exact match	54.5% (*p* = 0.075)	41.4% (*p* = 0.099)	47.47% (*p* = 0.929)
Plus I	16.8% (*p* = 0.063)	30.2% (*p* = 0.022) **	21.8% (*p* = 0.456)
Plus II	2.5% (*p* = 1.0)	3.9% (*p* = 0.222)	1.6% (*p* = 0.275)
Plus III	0% (*p* = 1.0)	0% (*p* = 1.0)	0.8% (*p* = 0.217)
Minus I	23.5% (*p* = 1.0)	21.05% (*p* = 0.365)	26.0% (*p* = 0.30)
Minus II	25.% (*p* = 1.0)	3.2% (*p* = 0.542)	2.1% (*p* = 0.587)

Tibia (*n* = 394)	<25% (*n* = 118)	<50% (*n* = 125)	>75% (*n* = 151)
Exact match	58.4% (*p* = 0.439)	50.4% (*p* = 0.231)	56.9% (*p* = 0.604)
Plus I	10.1% (*p* = 0.409)	13.6% (*p* = 0.746)	13.2% (*p* = 0.876)
Plus II	0.7% (*p* = 0.730)	0.8% (*p* = 0.283)	3.3% (*p* = 0.312)
Minus I	28.81% (*p* = 0.620)	28.8% (*p* = 0.544)	23.8% (*p* = 0.350)
Minus II	0% (*p* = 0.037) *	5,6% (*p* = 0.013) **	1.9% (*p* = 0.748)

**Table 5 jcm-12-01079-t005:** Kellgren–Lawrence score and adherence to planning; *p*-values from each Fischer’s exact test. Each experience group and match were tested against all other groups and matches. ** = significantly more often implanted than all other experience groups; K–L = Kellgren–Lawrence score.

Femur (*n* = 484)	K–L II (*n* = 12)	K–L III (*n* = 153)	K–L IV (*n* = 319)
Exact match	50% (*p* = 1.0)	43.13% (*p* = 0.327)	48.2% (*p* = 0.338)
Plus I	0% (*p* = 0.079)	30.1% (*p* = 0.30)**	21.94% (*p* = 0.118)
Plus II	8.3% (*p* = 0.244)	3.2% *p* = 0.336)	1.5% (*p* = 0.197)
Plus III	0% (*p* = 1.0)	0% (*p* = 1.0)	0.62% (*p* = 0.550)
Minus I	33.3% (*p* = 0.493)	20.9% (*p* = 0.305)	25.0% (*p* = 0.500)
Minus II	8.3% (*p* = 0.263)	1.9% (*p* = 0.760)	2.5% (*p* = 1.0)

Tibia (*n* = 370)	K–L II (*n* = 11)	K–L III (*n* = 115)	K–L IV (*n* = 244)
Exact match	54% (*p* = 1.0)	58.2% (*p* = 0.431)	53.6% (*p* = 0.443)
Plus I	0% (*p* = 0.372)	13.9% (*p* = 0.618)	12.7% (*p* = 1.0)
Plus II	0% (*p* = 1.0)	0.8% (*p* = 0.443)	2.4% (*p* = 0.430)
Minus I	45% (*p* = 0.176)	23.4% (*p* = 0.315)	27.86% (*p* = 0.711)
Minus II	0% (*p* = 1.0)	2.6% (*p* = 1.0)	2.8% (*p* = 1.0)

## Data Availability

Data from this research is available upon request to the corresponding author.

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
