# Peer review of "The Adherence of Digital Templating of Cemented Bicondylar Total Knee Arthroplasty Reveals Gender Differences"

_jcm, 2023, doi:10.3390/jcm12031079_

Round 1
Reviewer 1 Report
In this manuscript, the authors studied the effects of differences in gender, stage of arthrosis, and surgeon’s experience on the planning adherence for preoperative digital templating of total knee arthroplasty. This was a retrospective study with 521 patients who underwent a cemented primary bicondylar TKA and received either the S&N Genesis II CR/PS implant or the Journey II BCS/CR implant. The authors found that the exact planning adherence was 46.3% for the femur and 41.8% for the tibia. They found that there was no significant effect of surgeon experience on the planning adherence of digital templating. They also found that there were gender differences, with undersized implants being planned for men in a significant number of cases and oversized implants planned for women in a significant number of cases. Finally, the authors did not find any impact of the stage of arthrosis on the planning adherence on the tibial side but they did see a significant effect for the femoral side.
I have the following questions/comments for the authors:
(1.) The authors mention multiple different statistical tests in the Statistical analyses section on page 4 (lines 115 to 125). However, in the results section, it is not very clearly described how the statistical analysis was carried out for each data set. More specifically: (a) Please provide clear explanations for which specific tests/analyses were applied to each of the data sets in the results section. (b) In addition to clarifying which analysis method was applied to each data set, please also provide a brief explanation for why that particular statistical test was chosen in each case. (c) Some of the tests mentioned in the Statistical analyses section of page 4 receive no mention in the results section. For example, on lines 120 - 121 on page 4, the authors mention that the chi-square test and the fisher exact test were used to examine nominal variables. However, on line 206 of page 7, the authors mention that only the exact fisher’s test was applied. The chi-square test assumes that the sample is large whereas the fisher’s exact test is used for small sized samples. If the sample size was small enough only to apply the fisher’s exact test, then why do the authors state on page 4 that they applied the chi-square test? And if they did apply the chi-square test, then why were the results not reported? (d) Please provide more clarification on what the individual p-values mean in tables 4 and 5. What are you comparing against in each case?
(2.) Figure 1 on page 3 seems to be incomplete. Perhaps it is missing some arrows? The caption is also only a single word. Please include the complete figure with a more detailed caption.
(3.) The Discussion section summarizes the results of this study and provides comparisons with previous literature. It would be helpful if the authors could also discuss the impact/applications of these results. Why is it important to know the general levels of planning adherence and the specific effects of gender, arthrosis stage, and surgeon’s experience? How will this inform the next steps in the field of digital templating? The authors briefly address this in the conclusions section by stating that there is a further need for planning tools with higher accuracy (lines 282 and 283, page 9). Please could you expand upon this? What novel tools are currently being developed for this purpose? It would also be nice if the authors could provide their perspective/informed opinion on which tools have the most potential for improving accuracy.
(4.) The authors have not provided any information regarding author contributions, funding details, IRB statement, etc on pages 9 and 10. These sections currently have the journal guidelines. Please replace these with the relevant information for this particular manuscript.
Author Response
Dear Reviewer 1
Manuscript ID jcm-2169543
Thank you for giving us the opportunity to revise the manuscript “The adherence of digital templating of cemented bicondylar total knee arthroplasty reveals gender differences” submitted to the Journal of clinical medicine. The comments of the Reviewers have been carefully considered, and implemented as follows:
- The authors mention multiple different statistical tests in the Statistical analyses section on page 4 (lines 115 to 125). However, in the results section, it is not very clearly described how the statistical analysis was carried out for each data set. More specifically: (a) Please provide clear explanations for which specific tests/analyses were applied to each of the data sets in the results section. (b) In addition to clarifying which analysis method was applied to each data set, please also provide a brief explanation for why that particular statistical test was chosen in each case. (c) Some of the tests mentioned in the Statistical analyses section of page 4 receive no mention in the results section. For example, on lines 120 - 121 on page 4, the authors mention that the chi-square test and the fisher exact test were used to examine nominal variables. However, on line 206 of page 7, the authors mention that only the exact fisher’s test was applied. The chi-square test assumes that the sample is large whereas the fisher’s exact test is used for small sized samples. If the sample size was small enough only to apply the fisher’s exact test, then why do the authors state on page 4 that they applied the chi-square test? And if they did apply the chi-square test, then why were the results not reported? (d) Please provide more clarification on what the individual p-values mean in tables 4 and 5. What are you comparing against in each case?
L121 – L141
For patients demographics data were analyzed by mean, standard deviation and in percentage. For general planning adherence the planed size of femur and tibia was subtracted from the implanted size. 0 was the exact planning adherence. For planning adherence and sex, experience, and degree of arthrosis the nominal, dichotomous data were analyzed by Fischer’s exact test. This test was used, because of its independence from the sample size. The additional effect size phi (small .10; medium .30; large .50) was used to detect the clinical impact of a significant result. For planning adherence and sex each number from femur +III to -II was analyzed with sex. In addition, the exact matched, above and under sizes from femur and tibia were listed ordinally and analyzed through Mann-Whitney-U-Test to detect differences between sizes and sex and support Fischer’s exact test. The z-values of the Mann-Whitney-U-Test reflect the standard deviations. A z-value > -2 or 2 indicates that measured significances cannot be explained by theoretical random patterns. For planning adherence and experience and degree of arthrosis each experience level or degree of arthrosis was analyzed with the Fischer’s exact test. To support the results the Kruskal-Wallis-Test was used, because the evaluation of more than three independent samples is possible. The asymptomatic p and Kruskal-Wallis-H values were used in orientation to the significance level. K.-W-H values above 5,99 and asymptomatic p-values <0.05 show significance. Furthermore, linear regression with effect strength R2 (Cohen) (weak =0.02, middle =0.13 and high =0.26 variance clarification) was used to control the results. Through regression analyze it was investigated whether the degrees of experience or arthrosis have a linear influence on the planning adherence.
L217-L218 + L235 + L249
Additions in text of table:
2) (value of significance from Fischer’s exact test)
3) (value of significance from Fischer’s exact test)
4) p-values from each Fischer’s exact test. Each experience group and match was tested against all other groups and matches.
5) p-values from each Fischer’s exact test. Each experience group and match were tested against all other groups and matches.
L77-L87
(2.) Figure 1 on page 3 seems to be incomplete. Perhaps it is missing some arrows? The caption is also only a single word. Please include the complete figure with a more detailed caption.
Figure 1 Workflow. Data acquisition was consecutive. First, the examination of cases was done through the current patient recording system and, for control and additional data acquisition, through the archiving program of not digitally recorded documents. The examination of preoperative planed implants was carried out through the digital x-ray image viewer. The data collection then was analyzed for the study purposes shown in the figure.
(3)The Discussion section summarizes the results of this study and provides comparisons with previous literature. It would be helpful if the authors could also discuss the impact/applications of these results. Why is it important to know the general levels of planning adherence and the specific effects of gender, arthrosis stage, and surgeon’s experience? How will this inform the next steps in the field of digital templating? The authors briefly address this in the conclusions section by stating that there is a further need for planning tools with higher accuracy (lines 282 and 283, page 9). Please could you expand upon this? What novel tools are currently being developed for this purpose? It would also be nice if the authors could provide their perspective/informed opinion on which tools have the most potential for improving accuracy.
L260-L264
“Digital templating is an underestimated crucial feature which allows the surgeon to plan the best implant for the specific patient. It starts a preoperative preparation process that makes the decision for an implant size safer and more reliable. Better results of the outcome were shown when planning the implants before arthroplasty [16, 17].”
L276-277
“The influence of the experience of a surgeon in digital templating can be a decisive factor to succeed the arthroplasty [19, 20].”
L285-L287
“Digital templating and sex are discussed in literature. Other studies could not reveal any gender differences in relation to planning [10, 22]. Since there are gender differences in the outcome of arthroplasties [23, 24] a precise gender individual conformed planning could have a positive influence.“
L300-L304
“As the degree of arthrosis for a total arthroplasty is mostly in final stage a comparative analyze in literature was barley found [26]. Anyhow, most patients only show up for total knee arthroplasty when pain is out of control and the mobility is substantially limited. This is often in line with advanced stages of arthrosis [27]. The results of this study re-vealed that there is no difference between Kellgren-Lawrence stage II-IV, but overall, the adherence was poor.”
L306-L309
“Anyway, especially for extreme deformed joints a precise planed implant could be helpful for the surgery and might reduce the time of operation. As a result, in the future more 3D templating will be established. Accuracies of up to 90% and independence of different patient weight against 2D planning was demonstrated already [28, 29].”
L327-L333
“Currently low-dose CT 3D scans, MRI 3D scans or 3D reconstructions from 2D x-rays are in use to improve digital templating [6, 30, 31]. Another new technology are preoperative 3D-printed implants for planning the surgery [32]. This could be interesting for difficult cases. The greatest potential could be in 3D CT scans, where implants can be templated in all directions and more precise than in 2D-technique. Nevertheless, clinics still should not lose sight of the economic efficiency. At this point digital 2D-templating is still very important as a standard for the most of the surgeons.”
(4)The authors have not provided any information regarding author contributions, funding details, IRB statement, etc on pages 9 and 10. These sections currently have the journal guidelines. Please replace these with the relevant information for this particular manuscript.
Author Contributions: Conceptualization, writing—original draft preparation, J.K.; formal analysis, Methodology, J.T.; software, data curation, C.-D.P.; validation, writing—review and editing, M.T.; supervision, project administration, C.G.; All authors have read and agreed to the published version of the manuscript.
Funding:
This research received no external funding
Institutional Review Board Statement:
The study was conducted in accordance with the Declaration of Helsinki and approved by the Ethics Committee of RUB University (HDZ Bad Oeynhausen; file number: 2022-926; Date: 29.03.2022)
Informed Consent Statement:
Not applicable.
Data Availability Statement:
Data from this research is available upon request to the corresponding author.
Conflicts of Interest:
The authors declare no conflict of interest.
We hope that the changes implemented have improved the manuscript, and that it has now reached the standard necessary for formal acceptance in the Journal of clinical medicine.
We look forward to hearing from you
Your sincerely
The authors

Reviewer 2 Report
Thank you for the opportunity to comment on this document. The subject is interesting and requires further investigation. That being said, I do have a few concerns. 1º I would like to know what is the novelty of the article and its contribution to the knowledge of the scientific community. 2º The introduction should be expanded to provide a solid background of the main idea of the study. In addition, the authors must finish the introduction with an objective in accordance with the investigation that is intended. 3º Finally, the discussion should be improved with more evidence-based explanations of the results and comparing the finding with previous or relevant studies that show the importance of these studies and how the finding contributes to the scientific community. I think that a subject as important as this needs more bibliography.
Author Response
Dear Reviewer 2
Manuscript ID jcm-2169543
Thank you for giving us the opportunity to revise the manuscript “The adherence of digital templating of cemented bicondylar total knee arthroplasty reveals gender differences” submitted to the Journal of clinical medicine. The comments of the Reviewers have been carefully considered, and implemented as follows:
1º I would like to know what is the novelty of the article and its contribution to the knowledge of the scientific community.
L48-L51
The study of gender differences and surgeon’s experiences in digital templating for THA is well known. There is only little information about gender differences for total knee arthroplasty. To our knowledge there is no survey about sex differences, surgeons’s experience and stage of arthrosis in planning adherence all at once for bicondylar total knee arthroplasty.
2º The introduction should be expanded to provide a solid background of the main idea of the study. In addition, the authors must finish the introduction with an objective in accordance with the investigation that is intended.
L41-L51
Preoperative planning not only simplifies the determination of the exact implant size, it can improve the surgery procedure, postoperative range of motion and helps to restore patient specific biomechanics [6, 7]. A lot of studies in the last years were focused on planning adherence in total hip arthroplasty (THA) [6, 8, 9]. For example, Dammerer et al. 2022 showed a sex difference in templating the femur stem with better results for women. Likewise, Luger et al. 2022 revealed gender differences in planning adherence during total hip arthroplasty. For surgeons experience and digital planning several studies were conducted for THA. Some studies demonstrated differences with better planning adherence for more experienced surgeons, others could not find any differences [10-12]. An additional detailed view on the stage of arthrosis and its outcome for the digital planning adherence has not been conducted so far and remains unclear.
L57-L68
The study of gender differences and surgeon’s experiences in digital templating for THA is well known. There is only little information about gender differences for total knee arthroplasty. To our knowledge there is no survey about sex differences, surgeons’s experience and stage of arthrosis in planning adherence all at once for bicondylar total knee arthroplasty. In this study an investigation of planning adherence in relation to sex, experience of the surgeon and the degree of arthrosis for bicondylar total knee arthro-plasty was done. Thereby the exact planning, over and under planed sizes were analyzed in relation to gender experience of the surgeon and the degree of arthrosis. The aim of the study was the detection of repetitive significant differences in the planning adherence regarding gender, surgeons experience or stage of arthrosis to prevent possible occurring failures and improve the clinical work process.
3º Finally, the discussion should be improved with more evidence-based explanations of the results and comparing the finding with previous or relevant studies that show the importance of these studies and how the finding contributes to the scientific community.
L260-L264
“Digital templating is an underestimated crucial feature which allows the surgeon to plan the best implant for the specific patient. It starts a preoperative preparation process that makes the decision for an implant size safer and more reliable. Better results of the outcome were shown when planning the implants before arthroplasty [16, 17].”
L276-277
“The influence of the experience of a surgeon in digital templating can be a decisive factor to succeed the arthroplasty [19, 20].”
L285-L287
“Digital templating and sex are discussed in literature. Other studies could not reveal any gender differences in relation to planning [10, 22]. Since there are gender differences in the outcome of arthroplasties [23, 24] a precise gender individual conformed planning could have a positive influence.“
L300-L304
“As the degree of arthrosis for a total arthroplasty is mostly in final stage a comparative analyze in literature was barley found [26]. Anyhow, most patients only show up for total knee arthroplasty when pain is out of control and the mobility is substantially limited. This is often in line with advanced stages of arthrosis [27]. The results of this study re-vealed that there is no difference between Kellgren-Lawrence stage II-IV, but overall, the adherence was poor.”
L306-L309
“Anyway, especially for extreme deformed joints a precise planed implant could be helpful for the surgery and might reduce the time of operation. As a result, in the future more 3D templating will be established. Accuracies of up to 90% and independence of different patient weight against 2D planning was demonstrated already [28, 29].”
Actual bibliography was added from reference 19-35.
We hope that the changes implemented have improved the manuscript, and that it has now reached the standard necessary for formal acceptance in the Journal of clinical medicine.
We look forward to hearing from you
Your sincerely
The authors

Round 2
Reviewer 2 Report
Thanks for the corrections, no doubt now it has gained in quality